# Evaluating the Impact of Omega-3 Fatty Acid (Soloways^TM^) Supplementation on Lipid Profiles in Adults with PPARG Polymorphisms: A Randomized, Double-Blind, Placebo-Controlled Trial

**DOI:** 10.3390/nu16010097

**Published:** 2023-12-27

**Authors:** Evgeny Pokushalov, Andrey Ponomarenko, Sevda Bayramova, Claire Garcia, Inessa Pak, Evgenya Shrainer, Elena Voronina, Ekaterina Sokolova, Michael Johnson, Richard Miller

**Affiliations:** 1Center for New Medical Technologies, 630090 Novosibirsk, Russia; ponomarenko_av@cnmt.ru (A.P.); bayramova_sa@cnmt.ru (S.B.); inesspak@yandex.ru (I.P.); shrayner_ev@cnmt.ru (E.S.); 2Scientific Research Laboratory, Triangel Scientific, San Francisco, CA 94101, USA; info@triangelcomany.com (C.G.);; 3Institute of Chemical Biology and Fundamental Medicine, Siberian Branch of the Russian Academy of Sciences, 630090 Novosibirsk, Russia; voronina_l@mail.ru (E.V.); sokolovaea2608@gmail.com (E.S.)

**Keywords:** PPARG polymorphisms, omega-3 fatty acids, LDL-C, triglycerides, cardiovascular health, personalized medicine

## Abstract

Emerging evidence suggests that PPARG gene polymorphisms may influence lipid metabolism and cardiovascular risk, with omega-3 fatty acids proposed to modulate these effects. This study aims to assess the effects of fish oil supplementation on cardiovascular markers among adults with PPARG gene polymorphisms in a randomized, double-blind, placebo-controlled trial. A cohort of 102 patients with LDL-C 70–190 mg/dL was randomized to receive either 2000 mg of omega-3 fatty acids or a placebo daily for 90 days. In the omega-3 group with PPARG polymorphisms, LDL-C was reduced by 15.4% (95% CI: −19.8% to −11.0%), compared with a 2.6% decrease in the placebo group (95% CI: −4.1% to −1.1%; *p* < 0.01). In the omega-3 group without PPARG polymorphisms, LDL-C was reduced by 3.7% (95% CI: −6.9% to −0.6%), not significantly different from the placebo group’s reduction of 2.9% (95% CI: −5.1% to −0.8%; *p* = 0.28). The reduction in LDL-C was notably 11.7% greater in those with PPARG polymorphisms than in those without (95% CI: −19.3% to −4.0%; *p* < 0.01). Triglycerides decreased by 21.3% in omega-3 recipients with PPARG polymorphisms (95% CI: −26.5% to −16.2%; *p* < 0.01), with no significant changes in HDL-C, total cholesterol, or hsCRP levels in any groups. Minor allele frequencies and baseline characteristics were comparable, ensuring a balanced genetic representation. Omega-3 fatty acids significantly reduce LDL-C and triglycerides in carriers of PPARG polymorphisms, underlining the potential for genetic-driven personalization of cardiovascular interventions.

## 1. Introduction

Cardiovascular diseases (CVDs) remain a predominant cause of morbidity and mortality globally, underscoring the necessity for efficacious preventive and therapeutic strategies [1]. Dyslipidemia, characterized by elevated levels of low-density lipoprotein cholesterol (LDL-C) and triglycerides, alongside reduced high-density lipoprotein cholesterol (HDL-C) levels, plays a pivotal role in CVD pathogenesis [2].

Elevated cholesterol levels are identified as a primary modifiable risk factor for CVD. The management of cholesterol levels is crucial for mitigating the risks associated with CVD, especially in patients with Low-to-Moderate Cardiovascular Risk [2]. For these patients, non-pharmacological strategies, including dietary supplements, are often recommended over pharmacological interventions [2].

The Peroxisome Proliferator-Activated Receptor Gamma (PPARG) gene, a key regulator of fatty acid storage and glucose metabolism, has mutations linked to augmented cardiovascular risks [2]. The PPARG gene encodes the PPAR-gamma protein, a nuclear receptor that regulates the transcription of various genes involved in these processes. When activated, PPAR-gamma promotes the uptake and storage of fatty acids and glucose, playing a vital role in energy homeostasis, metabolic function, and cardiovascular health. Mutations in PPARG can disrupt these regulatory mechanisms, potentially leading to metabolic disorders and increased cardiovascular risk [3].

Omega-3 polyunsaturated fatty acids, primarily derived from marine oils, have demonstrated efficacy in enhancing cardiac function and attenuating inflammatory processes, thus representing a prospective modulatory therapeutic approach for patients harboring mutations in the PPARG gene [4,5]. These fatty acids can modulate the expression of PPARG, enhancing the transcription of genes involved in fatty acid oxidation and cholesterol transport, potentially leading to lowered LDL-C levels and reduced cardiovascular risk [6]. The anti-inflammatory properties of omega-3 fatty acids, coupled with PPAR-gamma’s role in inflammation modulation, further contribute to a potentially synergistic effect in mitigating cardiovascular risks [7].

Previous research suggests a nuanced interaction between omega-3 fatty acids and genetic variations at the PPARG locus, potentially affecting blood lipid profiles and, by extension, cardiovascular risk [8]. However, the superiority of fish oil supplementation, rich in omega-3 fatty acids, in improving cardiovascular markers, specifically in patients with PPARG gene mutations compared to a placebo, remains an uncharted domain.

This study aims to bridge this knowledge gap by conducting a randomized, double-blind, placebo-controlled superiority trial to ascertain the efficacy of fish oil supplementation in ameliorating cardiovascular markers among adults with confirmed PPARG gene polymorphisms.

## 2. Materials and Methods

This was a randomized, double-blind, parallel-group clinical trial that compared treatment with omega-3 fatty acids to placebo. The study protocol was approved by the local Ethics Committee and conducted in compliance with the protocol and in accordance with standard institutional operating procedures and the Declaration of Helsinki. All patients enrolled in the study provided written informed consent. The study was registered with ClinicalTrials.gov (NCT06154408).

### 2.1. Patient Population and Design

Patients were eligible based on the following criteria:

Inclusion criteria:Age between 40 and 75;LDL-C level between 70 and 190 mg/dL, confirmed in at least two sequential checks conducted within the last six months prior to signing the consent form.

Exclusion criteria:Personal history of cardiovascular disease or high risk (≥20%);Triglycerides (TG) ≥ 400 mg/dL;Body Mass Index ≥ 35 kg/m^2^;Assumption of lipid-lowering drugs or supplements affecting lipid metabolism within the last three months;Diabetes mellitus;Known severe or uncontrolled thyroid, liver, renal, or muscle diseases.

Patients were randomized to omega-3 fatty acids (*n* = 51) or placebo (*n* = 51) by a computer-generated random sequence. Neither the researchers randomizing the patients nor the patients themselves knew the treatments they were allocated before the randomization or during the study. Participants in each group took 2 capsules/day. Omega-3 fatty acids and placebo capsules were identical in appearance and matched for color coating, shape, and size. The active treatment, supplied by S.Lab (SOLOWAYS), Ltd., Novosibirsk, Russia, contained omega-3 fatty acids (1000 mg of fish oil, of which 600 mg were eicosapentaenoic acid and 300 mg docosahexaenoic acid) per capsule. The study lasted for 90 days. All participants were advised to adhere to a diet with a balanced intake of proteins, fats, and carbohydrates while maintaining their usual lifestyle and medication regimen. This guidance aimed to evaluate the effects of omega-3 Fatty Acid Supplementation in a realistic setting, reflecting typical dietary and lifestyle conditions. The consumption of supplements and placebos during the study was monitored by asking people to return the medication containers and through brief daily cell phone reminders for participants to take the supplements.

Participants had a fasting lipid panel, complete metabolic panel, and high-sensitivity *C*-reactive protein (hsCRP) measured at day 0 and day 90 of the study. LDL-C was calculated using the Friedewald equation.

After randomization, all patients provided DNA samples for SNP selection and genotyping. Following genotyping, patients exhibiting at least one of the following PPARG gene polymorphisms: rs10865710 (G allele), rs7649970 (T allele), rs1801282 (G allele), and rs3856806 (T allele) were allocated into a separate subgroup [2,9,10]. Consequently, four subgroups were established: 1. the group with omega-3 fatty acids and polymorphism, 2. the group with omega-3 fatty acids without polymorphisms, 3. the placebo group with polymorphism, and 4. the placebo group without polymorphisms (Figure 1).

In this study, S.Lab (SOLOWAYS), a pharmaceutical company, contributed solely by manufacturing the omega-3 fatty acid supplements used in the research. S.Lab (SOLOWAYS) did not participate in the design, execution, or financing of the experiment beyond providing the required supplements. The entire study was independently conducted by a research team from the Center for New Medical Technologies and the Scientific Research Laboratory at Triangel Scientific. This arrangement ensured that the study’s outcomes were not influenced by commercial interests, maintaining the integrity and independence of our research.

### 2.2. Study Endpoints

The primary endpoint was the percent change in LDL-C for omega-3 fatty acids compared with placebo among patients with PPARG polymorphism. Secondary endpoints included the percent change in hsCRP, high-density lipoprotein cholesterol (HDL-C), total cholesterol, and serum triglycerides between subgroups.

### 2.3. Sample Size Calculation and Statistical Power

For our randomized, placebo-controlled trial, the primary objective was to discern a 15% reduction in LDL-C levels in the omega-3 fatty acid group compared to the placebo group, with an anticipated average LDL-C level of 145 mg/dL. We considered a standard deviation of 15% in LDL-C level changes across both groups. The sample size calculation, incorporating a two-sided significance level of 5% and aiming for a power of 90%, was adjusted to account for the frequency of PPARG polymorphism. We initially estimated, based on the Hardy–Weinberg equilibrium and the minor allele frequency of PPARG, that 48 participants per group would be sufficient. To account for the PPARG polymorphism distribution in our target population and a potential dropout rate of 5%, we further adjusted the number. Consequently, we determined that enrolling 51 participants per group, inclusive of both polymorphic and non-polymorphic patients, would maintain the study’s statistical power. This adjustment brought the total sample size for the trial to 102 participants.

### 2.4. Statistical Analyses

The primary statistical analysis was centered on evaluating the efficacy of omega-3 fatty acids in reducing LDL-C levels compared to placebo. The analysis was conducted on an intention-to-treat basis, including all randomized patients who received at least one dose of the study medication and had at least one post-baseline efficacy evaluation. Percent changes in LDL-C levels from baseline to study end were calculated, and the comparison between the omega-3 fatty acid and placebo groups was performed using an independent *t*-test, assuming equal variances.

Secondary analyses compared changes in high-sensitivity *C*-reactive protein (hsCRP), high-density lipoprotein cholesterol (HDL-C), total cholesterol, and serum triglycerides between the subgroups. These were performed using a two-way ANOVA with treatment and genotype as factors, followed by post-hoc tests with Bonferroni correction for multiple comparisons to control the family-wise error rate.

To address the possibility of genotype-related treatment effects, interaction terms between the treatment group and genotype were included in the models. Subgroup analyses were pre-specified for patients with and without PPARG polymorphisms based on the presence of the selected single nucleotide polymorphisms (SNPs).

All tests were two-sided, with a significance level set at 0.05. Statistical power was maintained at 90%, accounting for a 5% dropout rate, resulting in a total sample size of 102 participants. Continuous variables were expressed as mean ± standard deviation (SD), and categorical variables were summarized with counts and percentages.

For the analysis of SNP data, the Hardy–Weinberg equilibrium was tested using a chi-square test for each SNP among the placebo group.

All statistical analyses were performed using SAS version 9.4 (SAS Institute, Cary, NC, USA), and figures were generated to visualize the distribution of changes and the treatment effects across subgroups.

This analytical approach was designed to ensure robust detection of treatment effects and interactions, providing a comprehensive understanding of the impact of omega-3 fatty acids on lipid metabolism in the context of genetic variability.

## 3. Results

A total of 102 patients were subjected to randomization. Among these, 99 successfully completed both the baseline and follow-up laboratory evaluations, as depicted in Figure 1. Adherence to the study regimen was high, with only four participants taking less than 70% of their assigned regimen (two from the omega-3 fatty acids group and two from the placebo group), and dietary habits remained unchanged overall during the study. The minor allele frequency for rs10865710 (G allele), rs7649970 (T allele), rs1801282 (G allele), and rs3856806 (T allele) was 0.201, 0.088, 0.108, and 0.127, respectively, and all SNPs were in Hardy–Weinberg equilibrium (*p* > 0.05).

The baseline characteristics of patients assigned to the omega-3 fatty acids and placebo groups displayed no significant differences in most parameters, indicating comparable groups at the study’s initiation. Subgroup analysis, however, did reveal significant differences in lipid profiles associated with PPARG polymorphisms. Specifically, patients with PPARG polymorphisms exhibited higher baseline levels of certain lipids.

The mean baseline LDL-C level was higher in patients with PPARG polymorphisms (135.3 ± 23.2 mg/dL) compared to those without (123.7 ± 19.3 mg/dL, *p* = 0.04 for the omega-3 fatty acids group and 139.2 ± 27.1 mg/dL vs. 119.9 ± 17.8 mg/dL, *p* = 0.02 for the placebo group).

Similarly, mean HDL-C levels differed (52.4 ± 13.5 mg/dL for those with polymorphisms vs. 61.9 ± 16.6 mg/dL for those without, *p* = 0.03 in the omega-3 fatty acids group and 51.5 ± 11.6 mg/dL vs. 59.2 ± 15.1 mg/dL, *p* = 0.05 in the placebo group).

Total cholesterol levels also showed significant differences (215 ± 25 mg/dL for patients with polymorphisms vs. 200 ± 22 mg/dL for those without, *p* = 0.05 in the omega-3 fatty acids group, and 220 ± 28 mg/dL vs. 198 ± 20 mg/dL, *p* = 0.03 in the placebo group).

Triglyceride levels were 160 ± 35 mg/dL for those with PPARG polymorphisms compared to 140 ± 25 mg/dL for those without (*p* = 0.03) in the omega-3 fatty acids group, and 165 ± 40 mg/dL vs. 135 ± 20 mg/dL (*p* = 0.02) in the placebo group.

The hsCRP levels did not differ significantly between the subgroups (2.5 ± 1.7 mg/L for patients with polymorphisms vs. 1.9 ± 1.3 mg/L for those without, *p* = 0.20 in the omega-3 fatty acids group; and 2.6 ± 1.8 mg/L vs. 1.8 ± 1.2 mg/L, *p* = 0.19 in the placebo group).

These findings are detailed in Table 1.

### 3.1. Primary Endpoint

Table 2 and Figure 2 illustrate the percentage change in LDL-C from baseline to day 90 in both the omega-3 fatty acid group and the placebo group. Among patients with PPARG polymorphisms, the omega-3 fatty acid group experienced an average LDL-C decrease of 15.4% from baseline (95% CI: −19.8% to −11.0%), which was significantly greater than the placebo group with PPARG polymorphism, where the change was −2.6% (95% CI: −4.1% to −1.1%). The difference in LDL-C reduction between the omega-3 fatty acids group with PPARG polymorphism and the placebo group with the same polymorphism was 12.8% (95% CI: −21.7% to −3.9%; *p* < 0.01).

In patients without PPARG polymorphisms, the omega-3 fatty acid group showed a less pronounced mean LDL-C decrease from baseline, at 3.7% (95% CI: −6.9% to −0.6%). However, the LDL-C reduction in the placebo group without PPARG polymorphisms was −2.9% (95% CI: −5.1% to −0.8%), resulting in no significant difference between the omega-3 fatty acids group without PPARG polymorphisms and the placebo group without the polymorphisms (*p* = 0.28). Notably, the reduction in LDL-C among patients treated with omega-3 fatty acids was 11.7% (95% CI: −19.3% to −4.0%) greater in those with PPARG polymorphisms than in those without (*p* < 0.01).

### 3.2. Secondary Endpoints

Table 2 illustrates the percent change from baseline to day 90 in other lipid biomarkers (total cholesterol, HDL-C, triglycerides) and hsCRP.

The reduction in serum triglycerides among patients with PPARG polymorphisms was greater with omega-3 fatty acid treatment than with placebo, at 21.3% (95% CI: −26.5% to −16.2%) for omega-3 fatty acids vs. −1.9% (95% CI: −4.7% to 0.9%) for placebo (*p* < 0.01) (Figure 3). The reduction in patients treated with omega-3 fatty acids was greater in those with PPARG polymorphisms than those without, at 12.8% (95% CI: −22.2% to −3.4%; *p* < 0.01).

No significant differences were observed in total cholesterol, HDL-C, and hsCRP levels among patients with PPARG polymorphisms when comparing omega-3 fatty acids with placebo; changes were −4.8% (95% CI: −13.6% to 4.1%) for total cholesterol, 3.4% (95% CI: −5.4% to 12.3%) for HDL-C, and −1.9% (95% CI: −46.3% to 42.5%) for hsCRP (*p* > 0.05 for all). Similarly, there were no significant differences in total cholesterol, HDL-C, and hsCRP levels when comparing patients treated with omega-3 fatty acids with and without PPARG polymorphisms (*p* > 0.05 for all). Also, no significant gender differences in LDL-C reduction among participants, irrespective of their PPARG polymorphism status.

## 4. Discussion

In the present study, we successfully replicated the interaction of genetic variants at PPARG with omega-3 fatty acids on blood lipids. The principal outcomes of this randomized, double-blind, parallel-group clinical trial include: (1) among patients with PPARG polymorphisms, elevated cholesterol levels, and low-to-moderate cardiovascular risk, the intake of omega-3 fatty acids significantly reduced LDL-C and serum triglycerides by the end of 3 months; and (2) patients with PPARG polymorphisms exhibited a more pronounced reduction in LDL-C, and serum triglycerides in response to omega-3 fatty acid supplementation compared to similar patients without the PPARG polymorphisms.

The results of this study provide compelling evidence for the differential impact of omega-3 fatty acid supplementation on lipid profiles in the context of PPARG polymorphisms. Consistent with previous research indicating the role of PPARG in lipid metabolism, our findings suggest a modulatory effect of genetic variation on the efficacy of dietary interventions [11].

PPARG genes have been extensively studied for their role in glucose homeostasis and insulin sensitivity. Specifically, PPAR-gamma activation, a result of PPARG gene expression, enhances glucose tolerance and insulin sensitivity in both diabetes mellitus patients and animal models of insulin resistance. This activation leads to marked improvements in insulin and glucose parameters due to enhanced whole-body insulin sensitivity, though the precise mechanisms are not fully understood [12,13,14,15]. Additionally, the Pro12Ala polymorphism in the PPAR-gamma2 gene variant is linked to insulin sensitivity in Brazilian diabetes mellitus patients, suggesting a genetic basis for varying metabolic responses. This link is particularly notable among Brazilian Caucasians, with individuals carrying the Ala12 allele of PPAR-gamma2 showing higher insulin sensitivity than Pro12 allele carriers [16].

Subsequent studies have identified specific polymorphisms in the PPARG promoter, notably rs10865710 and rs3856806, which are significantly associated with glucose levels in diabetes mellitus patients. The rs10865710 polymorphism is also correlated with the severity of coronary artery disease (CAD), potentially influenced by hyperlipidemia and hyperglycemia. Diabetes mellitus patients with the G allele of rs10865710 demonstrate higher levels of glucose, triglycerides, and other markers compared to non-carriers. Similarly, T allele carriers of rs3856806 show elevated glucose levels. Notably, the rs10865710 polymorphism is significantly related to CAD severity, likely due to its effects on lipid and glucose metabolism [17]. These findings highlight a complex interplay between PPARG polymorphisms and metabolic disorders, such as hypercholesterolemia and hypertriglyceridemia, potentially increasing CAD susceptibility.

Thus, the broader implications of PPARG polymorphisms on metabolic health are not limited to lipid metabolism but also include significant roles in glucose regulation and insulin sensitivity. Understanding these relationships is crucial for developing tailored dietary and therapeutic strategies for individuals with specific genetic profiles, especially in managing metabolic syndrome and diabetes.

In our cohort, the substantial decrease in LDL-C levels by 15.4% among those with PPARG polymorphisms receiving omega-3 fatty acids is significantly higher than the 2.6% reduction observed in the placebo group. This effect size is comparable to outcomes observed in earlier trials involving omega-3 supplementation for dyslipidemia management [18]. Notably, our study extends these findings by demonstrating that genetic variation can influence therapeutic outcomes, aligning with the concept of nutrigenomics, where nutrition is tailored to an individual’s genetic profile [19].

The lack of significant LDL-C reduction in patients without PPARG polymorphisms receiving omega-3 fatty acids underscores the necessity of genetic screening before dietary interventions. This observation is in line with the personalized medicine approach, which has been gaining traction in the management of cardiovascular diseases (CVD) and highlights the importance of individual genetic makeup in treatment response [20].

The secondary endpoint analysis revealed that omega-3 fatty acids also led to a marked reduction in serum triglycerides by 13.3% in the genetically predisposed group, which supports the lipid-lowering capacity of omega-3 fatty acids as documented in a meta-analysis by Eslick et al. [21]. However, in contrast to some studies that report improvements in HDL-C and total cholesterol, our results did not show significant changes in these markers [22]. This discrepancy might be attributed to the specific genetic makeup of our study population, suggesting that the interplay between genetics and lipid outcomes warrants further investigation.

The absence of significant differences in hsCRP levels indicates that the anti-inflammatory effects of omega-3 fatty acids may not be as pronounced in a short-term intervention or may require a longer duration to manifest, as suggested by other long-term studies [23].

Furthermore, the differential response to omega-3 fatty acids in our study raises questions about the potential role of other genetic factors in lipid metabolism. For example, polymorphisms in other genes related to lipid transport and metabolism, such as APOE, LDLR, and PCSK9, might also influence the response to dietary interventions. Previous studies have indicated that variations in these genes can significantly affect lipid levels and cardiovascular risk [24,25]. Therefore, comprehensive genetic profiling, including these genes, could provide a more holistic understanding of individual responses to omega-3 supplementation and enable more effective personalized dietary recommendations.

Another aspect that warrants further exploration is the role of dietary patterns and lifestyle factors in conjunction with genetic predispositions. While our study focused on the supplementation of omega-3 fatty acids, the interaction of these supplements with overall dietary habits, physical activity, and other lifestyle choices could significantly influence lipid profiles and cardiovascular health. Studies have shown that dietary patterns, such as the Mediterranean diet, which is rich in natural sources of omega-3, along with regular physical activity, have a synergistic effect in improving cardiovascular health [26,27]. Investigating these interactions in genetically diverse populations could provide valuable insights into comprehensive strategies for managing dyslipidemia and reducing cardiovascular risk.

In addition to the outcomes observed in this study, the inclusion of biomarkers for pre-inflammatory and preclinical atherosclerotic states, such as intima-media-thickness of carotid arteries, IL-6, TGF-b1, FGF, and ox-LDL, presents a compelling avenue for future research [28,29]. The relevance of carotid intima-media thickness as a surrogate marker for generalized atherosclerosis is well-established, and its association with inflammatory markers like *C*-reactive protein and oxidized LDL highlights its potential in cardiovascular disease research. These markers are crucial in understanding the progression of atherosclerosis and inflammation. Further investigation into these biomarkers in the context of omega-3 supplementation in individuals with PPARG polymorphisms could offer deeper insights into personalized cardiovascular health interventions.

The strengths of our study include its randomized, double-blind, placebo-controlled design and stringent adherence monitoring, which reinforce the validity of our findings. However, limitations exist, including the relatively short duration of the study and the specific genetic focus, which may not be generalizable to all populations with different genetic backgrounds.

Future research should explore the long-term effects of omega-3 fatty acid supplementation across diverse genetic landscapes to establish broader implications for CVD risk management. Additionally, investigations into the molecular mechanisms by which PPARG polymorphisms modulate lipid metabolism could offer insights into the development of targeted therapies [30].

## 5. Conclusions

Our study provides robust evidence supporting the modulatory role of PPARG polymorphisms in lipid metabolism in response to omega-3 fatty acid supplementation. The data obtained from this randomized, double-blind, parallel-group clinical trial unequivocally demonstrate that patients with PPARG polymorphisms, who are at low-to-moderate cardiovascular risk and present with elevated cholesterol levels, benefit significantly from the introduction of omega-3 fatty acids into their diet. These benefits are manifested in substantial reductions in LDL-C, total cholesterol, and serum triglycerides over a three-month period.

Importantly, the findings highlight the potential for personalized nutrition strategies, suggesting that genetic profiling could be an invaluable tool in optimizing lipid-lowering interventions. Patients carrying PPARG polymorphisms showed greater improvements in lipid profiles, indicating that genetic predispositions should be considered when prescribing omega-3 fatty acid supplementation. This could lead to more tailored and, consequently, more effective approaches to managing dyslipidemia and reducing cardiovascular risk.

Future research should aim to expand on these findings by exploring the long-term effects of omega-3 supplementation in genetically diverse populations and elucidating the molecular mechanisms underlying the interaction between PPARG variants and lipid metabolism. Our study lays the groundwork for a new paradigm in personalized medicine, where genetic information guides clinical decisions, leading to improved patient outcomes.

## Figures and Tables

**Figure 1 nutrients-16-00097-f001:**
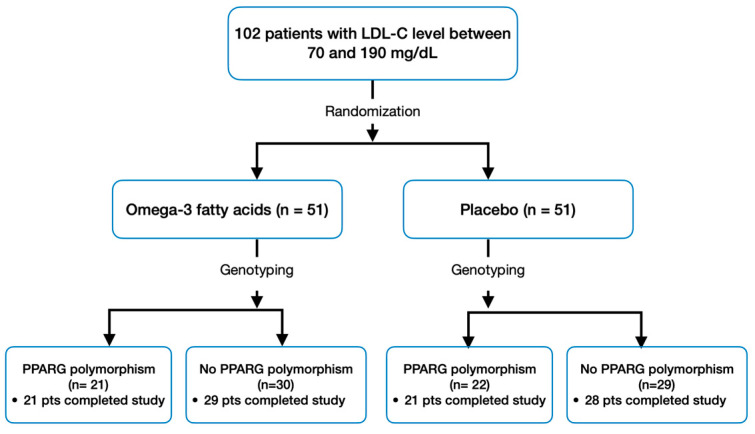
Study Design and Patient Flow.

**Figure 2 nutrients-16-00097-f002:**
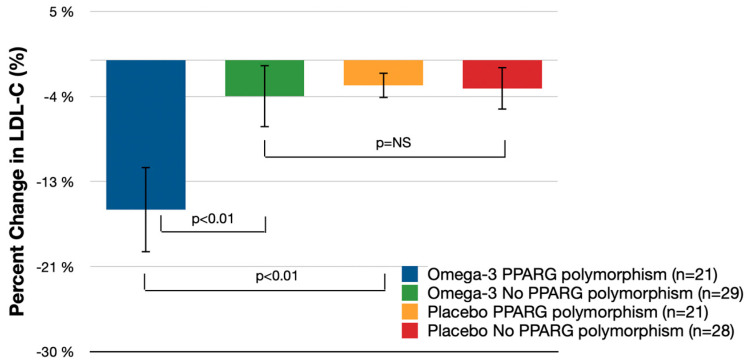
Mean Percent LDL-C Change (% Change from Baseline (95% CI); independent *t*-tests); NS—not significant.

**Figure 3 nutrients-16-00097-f003:**
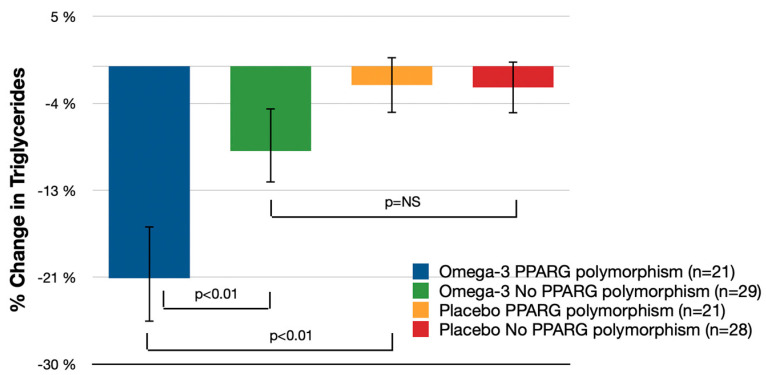
Mean Percent Serum Triglycerides Change (% Change from Baseline (95% CI); independent *t*-tests); NS—not significant.

**Table 1 nutrients-16-00097-t001:** Baseline population characteristics by genetic variants at PPARG (mean ± SD, *t*-tests, chi-square tests).

	Omega-3 Fatty Acids (*n* = 50)		*p*-Value	Placebo (*n* = 49)		*p*-Value
	PPARG Polymorphism (*n* = 21)	No PPARG Polymorphism (*n* = 29)		PPARG Polymorphism (*n* = 21)	No PPARG Polymorphism (*n* = 28)	
Age, y	62.7 ± 6.0	63.5 ± 10.1	*p* = 0.76	60.8 ± 10.4	59.5 ± 10.2	*p* = 0.69
Women, %	57.1	55.2	*p* = 0.71	54.5	60.7	*p* = 0.28
Body mass index, kg/m^2^	28.5 ± 3.2	27.9 ± 3.0	*p* = 0.68	28.7 ± 3.4	27.8 ± 2.9	*p* = 0.70
10-y risk ASCVD risk, %	7.9	7.5	*p* = 0.75	8.2	7.3	*p* = 0.61
Total cholesterol, mg/dL	215 ± 25	200 ± 22	*p* = 0.05	220 ± 28	198 ± 20	*p* = 0.03
LDL-C, mg/dL	135.3 ± 23.2	123.7 ± 19.3	*p* = 0.04	139.2 ± 27.1	119.9 ± 17.8	*p* = 0.02
HDL-C, mg/dL	52.4 ± 13.5	61.9 ± 16.6	*p* = 0.03	51.5 ± 11.6	59.2 ± 15.1	*p* = 0.05
Triglycerides, mg/dL	160 ± 35	140 ± 25	*p* = 0.03	165 ± 40	135 ± 20	*p* = 0.02
hsCRP, mg/L	2.5 ± 1.7	1.9 ± 1.3	*p* = 0.20	2.6 ± 1.8	1.8 ± 1.2	*p* = 0.19

**Table 2 nutrients-16-00097-t002:** Primary and Secondary Endpoints (independent *t*-tests, two-way ANOVA).

	Omega-3 Fatty Acids (*n* = 50) (% Change from Baseline (95% CI))		Omega-3 Fatty Acids PPARG Polymorphism vs. Omega-3 Fatty Acids No PPARG Polymorphism		Placebo (*n* = 49) (% Change From Baseline (95% CI))		Omega-3 Fatty Acids PPARG Polymorphism vs. Placebo PPARG Polymorphism		Omega-3 Fatty Acids No PPARG Polymorphism vs. Placebo No PPARG Polymorphism	
	PPARG Polymorphism (*n* = 21)	No PPARG Polymorphism (*n* = 29)	% Difference (95% CI)	*p*-Value	PPARG Polymorphism (*n* = 21)	No PPARG Polymorphism (*n* = 28)	% Difference (95% CI)	*p*-Value	% Difference (95% CI)	*p*-Value
LDL-C	−15.4% (−19.8%, −11.0%)	−3.7% (−6.9%, −0.6%)	−11.6% (−19.3%, −4.0%)	*p* < 0.01	−2.6% (−4.1%, −1.1%)	−2.9% (−5.1%, −0.8%)	−12.8% (−21.7%, −3.9%)	*p* < 0.01	0.2% (−5.0%, 5.4%)	*p* = 0.28
Total cholesterol	−6.2% (−10.9%, −1.5%)	−1.4% (−3.4%, 0.7%)	−4.9% (−8.6%, −1.1%)	*p* = 0.23	−1.4% (−3.3%, 0.4%)	−1.5% (−3.8%, 0.8%)	−4.8% (−13.6%, 4.1%)	*p* = 0.29	0.13% (−5.06%, 5.32%)	*p* = 0.61
HDL-C	6.7% (2.1%, 11.3%)	4.4% (0.1%, 8.6%)	2.3% (−3.8%, 8.4%)	*p* = 0.21	3.23% (−1.3%, 7.3%)	5.0% (0.6%, 9.5%)	3.4% (−5.4%, 12.3%)	*p* = 0.18	−0.7% (−5.9%, 4.5%)	*p* = 0.14
Triglycerides	−21.3% (−26.5%, −16.2%)	−8.5% (−12.8%, −4.3%)	−12.8% (−22.2%, 3.4%)	*p* < 0.01	−1.9% (−4.7%, 0.9%)	−2.1% (−4.8%, 0.5%)	−13.4% (−22.1%, −4.8%)	*p* < 0.01	−7.4% (−14.0%, −0.8%)	*p* = 0.21
hsCRP	−5.8% (−32.5%, 20.9%)	−2.1%(−24.1%, 19.9%)	−3.7% (−8.4%, 1.0%)	*p* = 0.56	−3.9%(−21.6%, 13.8%)	−6.1% (−23.9%, 11.7%)	−1.9% (−46.3%, 42.5%)	*p* = 0.82	−4.0% (−32.3%, 24.3%)	*p* = 0.72

## Data Availability

The data presented in this study are available on request from the corresponding author. The data are not publicly available as the participants did not consent to their data being shared publicly.

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
