# Peer review of "Evaluating the Impact of Omega-3 Fatty Acid (SolowaysTM) Supplementation on Lipid Profiles in Adults with PPARG Polymorphisms: A Randomized, Double-Blind, Placebo-Controlled Trial"

_nutrients, 2023, doi:10.3390/nu16010097_

Round 1

Reviewer 1 Report

Comments and Suggestions for Authors

This manuscript presents an interesting topic of fish oil supplementation effects on cardiovascular (CV) risk via markers among adults with PPARG polymorphisms, reducing LDL-C and triglycerides in carriers of PPARG polymorphisms, underlining the potential for genetic-driven personalization of CV interventions.

The paper is well written, the references are up-to-date and appropriate and the language seems correct throughout the text.

However, a better design between the group patients concerning a pre-inflammatory and preclinical atherosclerotic status associated with biomarkers as for example intima-media-thickness of carotid arteries, IL-6, TGF-b1, FGF, ox-LDL could be made. The authors are encouraged to add 3-4 sentences-lines with relative knowledge from the literature in order to better describe this important clinical model.

Author Response

Dear Reviewer,

Thank you very much for your constructive and insightful feedback on our manuscript. We appreciate your comments regarding the interesting topic and the overall quality of the paper, including the up-to-date references and appropriate language.

In response to your suggestion for a more detailed design concerning pre-inflammatory and preclinical atherosclerotic status associated with biomarkers, we have added a new paragraph in the discussion section. This paragraph incorporates the suggested biomarkers such as intima-media-thickness of carotid arteries, IL-6, TGF-b1, FGF, and ox-LDL, with relevant literature references to enhance the description of this important clinical model. We believe this addition will significantly strengthen our manuscript and thank you for guiding us to this improvement.

Sincerely,
Evgeny Pokushalov

Reviewer 2 Report

Comments and Suggestions for Authors

General comments:

The study should be very valuable reading for Nutrients readers: cardiovascular risk reduction with omega-3 dietary replacement is possible in PPARG polymorphisms. The authors investigatedthe effects of fish oil supplementation on cardiovascular markers among adults with PPARG gene polymorphisms in a randomized, double-blind, placebo-controlled trial. The results suggested that patients with PPARG polymorphisms, who are at low-to-moderate cardiovascular risk and present with elevated cholesterol levels, benefit significantly from the introduction of omega-3 fatty acids into their diet.

It was considered that the study was well structured, and the result included novelty; however, several points should be addressed to improve the manuscript.

1. Why was obesity defined as above 32kg/m2 in the exclusion criteria? Based on BMI 25-29.9 oveweight, over 30.0 obese. Why was 32kg/m2 the cutoff in your study? 

2. What was the diet of the participants, how much fatty, cholesterol-rich food did they eat?

3. Have gender differences been investigated, are gender differences in LDL-C reduction expected?

4.Was there a difference in insulin sensitivity in the study population at baseline and did it change with omega-3 treatment?

5. A paragraph at the end of the discussion on how to supplement with omega-3-acid would be recommended to go along with the recommended dosage for women, men, menopause, normal body weight vs. obesity.

6. Tables and figure captions in the results section should include the statistical tests carried out.

Author Response

Dear Reviewer,

Thank you for your valuable feedback on our manuscript. We appreciate your insights and have addressed your concerns as follows:

  1. We acknowledge the error in defining obesity in our exclusion criteria. The correct criterion is a Body Mass Index ≥ 35 kg/m2, and we have amended this in our manuscript.

  2. Participants were advised to maintain a balanced diet in proteins, fats, and carbohydrates, without specific monitoring or assessment of their diet. We have clarified this in the manuscript.

  3. Our study did not reveal significant gender-based differences in LDL-C reduction. We have included this observation in the manuscript.

  4. Insulin sensitivity changes were not evaluated in this study, but we recognize it as a valuable aspect for future research.

  5. While we appreciate your suggestion on omega-3-acid supplementation recommendations, our study's scope was specific to the effects of Omega-3 Fatty Acid Supplementation on lipid metabolism in PPARG polymorphisms, and did not extend to dosage recommendations for various demographic groups.

  6. We have updated all tables and figures in the results section to include the statistical tests used.

Thank you again for your constructive comments, which have helped enhance our manuscript.

Sincerely,

Evgeny Pokushalov